# Heart Failure and Arrhythmias: Circadian and Epigenetic Interplay in Myocardial Electrophysiology

**DOI:** 10.3390/ijms26062728

**Published:** 2025-03-18

**Authors:** Chen Zhu, Shuang Li, Henggui Zhang

**Affiliations:** 1Key Laboratory of Medical Electrophysiology, Ministry of Education and Medical Electrophysiological Key Laboratory of Sichuan Province, (Collaborative Innovation Center for Prevention of Cardiovascular Diseases), Institute of Cardiovascular Research, Southwest Medical University, Luzhou 646099, China; 20234099120024@stu.swmu.edu.cn (C.Z.); 20214099120003@stu.swmu.edu.cn (S.L.); 2Beijing Academy of Artificial Intelligence, Beijing 100084, China; 3Biological Physics Group, Department of Physics and Astronomy, The University of Manchester, Manchester M13 9PL, UK

**Keywords:** circadian clock, circadian proteins, heart failure, epigenetic regulations

## Abstract

Emerging evidence underscores the impact of circadian rhythms on cardiovascular processes, particularly in conditions such as hypertension, myocardial infarction, and heart failure, where circadian rhythm disruptions are linked to disease progression and adverse clinical outcomes. Circadian clock proteins are intricately linked to myocardial electrophysiological remodeling and epigenetic pathways associated with arrhythmias in heart failure. In the context of heart failure, circadian clock dysregulation leads to electrophysiological remodeling in the cardiomyocytes, which can precipitate life-threatening arrhythmias such as ventricular tachycardia (VT) and ventricular fibrillation (VF). This dysregulation may be influenced by environmental factors, such as diet and exercise, as well as genetic factors. Moreover, epigenetic modifications in heart failure have been implicated in the regulation of genes involved in cardiac hypertrophy, fibrosis, and inflammation. The interplay between circadian clock proteins, myocardial electrophysiological remodeling, and epigenetic pathways in heart failure-related arrhythmias is complex and multifaceted. Further research is needed to elucidate how these processes interact and contribute to the development of arrhythmias in heart failure patients. This review aims to explore the connections between circadian rhythms, myocardial electrophysiology, and arrhythmias related to heart failure, with the goal of identifying potential therapeutic targets and interventions that may counteract the adverse effects of circadian disruptions on cardiovascular health.

## 1. Introduction

Heart failure is associated with various cardiac conditions, including ventricular arrhythmias [1]. In the United States, nearly 5 million individuals are affected by heart failure, with an annual increase of approximately 550,000 new cases [2]. Over the past 30 years, this prevalence has surged by 500%. Similarly, in China, the prevalence of heart failure is around 0.9%, translating to an estimated 5.85 million cases [3]. The incidence of heart failure increases with age, affecting approximately 10% of individuals over 75 years old, thereby representing a significant public health challenge [4]. The 2022 European Society of Cardiology guidelines on the treatment and prevention of ventricular arrhythmias and sudden cardiac death emphasize the high prevalence of ventricular arrhythmias in most heart failure patients [5].

Ventricular arrhythmias often exacerbate with the progression of heart failure, creating a reciprocal relationship that ultimately contributes to the overall deterioration of the condition, potentially leading to sudden cardiac death (SCD) [6]. The predictive value of ventricular arrhythmias in SCD among heart failure patients is not entirely clear. However, left ventricular dysfunction and decreased left ventricular ejection fraction are recognized as independent risk factors for SCD [5]. Epidemiological studies show SCD rate among heart failure patients in China is about 13% [7]. Extrapolating from these figures suggests that 600,000 heart failure patients die from SCD annually in China, with 50% of these deaths attributed to malignant arrhythmias [8].

Ventricular arrhythmias (VAs) are common clinical complications in patients with chronic heart failure and significantly impact morbidity and mortality [9]. Among various forms of ventricular arrhythmias, premature ventricular contractions (PVCs) and ventricular tachycardia (VT) are the most frequently encountered in clinical practice. The pathogenesis of ventricular arrhythmias primarily involves abnormalities in automaticity, reentry, and aberrant depolarization activity (early or delayed after-depolarizations) [10,11]. Clinically, VT is classified according to hemodynamic stability into stable and unstable forms. Stable VT is typically asymptomatic or presents with mild symptoms, whereas unstable VT can manifest severe clinical symptoms such as syncope, cardiac arrest, or even sudden cardiac death [12]. Additionally, VT episodes can be categorized based on duration into non-sustained VT (lasting less than 30 s) and sustained VT (lasting more than 30 s or requiring urgent termination due to hemodynamic compromise). Chronic heart failure patients often exhibit reentrant VT involving the His–Purkinje conduction system and ventricular myocardium, and in some cases, bundle branch reentrant VT, particularly in patients with dilated cardiomyopathy [13], and in some cases, bundle branch reentrant VT, particularly in patients with dilated cardiomyopathy [14]. The fundamental mechanisms underlying ventricular arrhythmias in chronic heart failure include disturbances in automaticity, reentry circuits [15], and abnormal depolarization processes (early or delayed afterdepolarizations), leading to complex arrhythmic events [16].

Normal cardiac function relies on the heart’s electrophysiological properties, including the generation of action potentials (APs) in cardiac cells and the conduction of these APs between cells. However, many cardiac diseases, such as arrhythmia, myocardial infarction, and heart failure, are associated with disruptions in cardiac electrophysiological properties, leading to altered AP characteristics and conduction patterns [17,18].

Recently, scientists have discovered a natural circadian rhythm system in the heart known as the “circadian clock” [19]. The recent review from Sandra Crnko et al. [20] shows that the circadian clock plays a crucial role in regulating numerous cardiovascular functions, such as endothelial activity, thrombus formation, blood pressure, and heart rate. Alteration of these 24-h rhythms contributes to the onset of cardiovascular diseases, including heart failure, myocardial infarction, and arrhythmias, by altering the normal circadian rhythm and contributing to sleep disturbances. Moreover, this alteration affects the development, risk factors, incidence, and outcomes associated with cardiovascular disease. This system significantly impacts cardiac electrophysiological properties. The circadian clock composes a network of proteins that exhibit varying expression and activity levels throughout the day, thereby regulating various physiological processes in the body.

Epigenetic pathways play a crucial role in mediating heart failure-related arrhythmias [21]. These pathways regulate gene expression and can be influenced by external factors such as diet, exercise, and stress [21]. Understanding the epigenetic mechanisms underlying heart failure-related arrhythmias may lead to the development of new treatments and prevention strategies to improve patient outcomes. Additionally, epigenetic profiling can help identify individuals at risk of developing arrhythmias related to heart failure, allowing for early intervention and management.

The purpose of this review is to comprehensively examine and elucidate the intricate molecular mechanisms by which circadian clock proteins regulate myocardial electrophysiological remodeling through epigenetic pathways. Specifically, this review aims to understand how these molecular processes contribute to the development of heart failure-related arrhythmias. It will cover several key areas: the role of circadian clock proteins in the heart, their integration into myocardial electrophysiological remodeling, the involvement of epigenetic pathways in arrhythmias associated with heart failure, and the clinical implications and therapeutic potential of these findings. By exploring the connections between circadian rhythms, myocardial electrophysiology, and heart failure-related arrhythmias, this review aims to provide insights into the potential therapeutic targets and interventions that may mitigate the adverse effects of circadian disruptions on cardiovascular health.

## 2. The Circadian Clock’s Roles in Cardiac Electrophysiology

### 2.1. Circadian Rhythms in Myocardial Cells

#### 2.1.1. Overview of Circadian Rhythms

The circadian rhythm, or circadian clock, is an evolutionary adaptation that enables organisms to synchronize their physiological processes with environmental factors such as light, temperature, and humidity, typically operating on a cycle of approximately 24 h [22]. This rhythm is primarily regulated by core clock genes, which include brain and muscle ARNT-like protein 1 (*BMAL1*), circadian locomotor output cycles kaput (*CLOCK*), period (*PER*), and cryptochrome (*CRY*) [23]. These genes form a complex feedback loop where BMAL1 and CLOCK act as positive regulatory factors, while PER and CRY serve as negative regulators. This feedback loop, along with additional regulation by factors such as receptor-related orphan receptor (ROR) and nuclear receptor subfamily 1 group D member 1 (Rev-Erbα), maintains the circadian rhythm’s roughly 24-h cycle [24,25]. The proteins BMAL1 and CLOCK form a complex that acts as a positive regulatory factor. This complex binds to specific DNA regions called E-boxes to promote the transcription of key circadian genes, including *PER* (Period) and *CRY*. The *PER* and *CRY* genes, once transcribed and translated into proteins, gradually accumulate in the cytoplasm. After reaching a threshold level, these proteins enter the nucleus and inhibit the activity of the BMAL1-CLOCK complex. This inhibition reduces the transcription of their own genes (*PER* and *CRY*), thus lowering PER and CRY levels over time. As the levels drop, BMAL1 and CLOCK activity resumes, starting the cycle again. This core feedback loop is fine-tuned by additional regulatory proteins, primarily ROR (receptor-related orphan receptor) and Rev-Erbα. ROR acts to activate *BMAL1* expression, while Rev-Erbα represses it. By balancing each other, *ROR* and Rev-Erbα ensure the stability and precision of the circadian cycle. Together, this system creates a self-sustaining cycle where gene expression rises and falls over approximately 24 h. These oscillations in gene expression and protein levels set the body’s circadian rhythms, coordinating physiological processes with the day–night cycle, including cardiovascular functions.

#### 2.1.2. Role of Circadian Rhythms in the Heart

The cardiovascular system exhibits clear circadian rhythms in physiological characteristics and electromechanical activity [26]. For instance, parameters like heart rate, blood pressure, and myocardial contractility fluctuate throughout the day, peaking in the afternoon and reaching their lowest levels at night [27]. These variations are not solely controlled by the autonomic nervous system but are also influenced by the central circadian clock, which coordinates the peripheral clocks in blood vessels and cardiac cells [20,28]. The circadian clock in the heart regulates the rhythmicity and contractility of myocardial cells, ensuring that the heart’s function aligns with the body’s overall circadian rhythm [29]. The main components of the circadian clock in the heart include the circadian oscillator, circadian input pathways, circadian output pathways, and circadian regulatory factors (Figure 1).

The circadian oscillator is a central component of the circadian clock that generates periodic changes in gene expression and protein levels, ultimately regulating various physiological processes in the body [23]. This core oscillator consists of transcriptional–translational feedback loops involving several genes and proteins, including BAML1, Clock, Per2, CRY, Rev-erbα, and ROR [23]. In the heart, these feedback loops do not merely oscillate in isolation—they are tightly coupled with both systemic signals (e.g., hormones, autonomic inputs) and local factors (e.g., metabolic demands) that fine-tune cardiac physiology across the 24-h cycle. Circadian input pathways are signals from the external environment that entrain or reset the circadian clock [30]. For the heart, the most important input pathway is light, which is detected by cells in the retina called specialized photoreceptors. Light information is transmitted to the SCN (suprachiasmatic nucleus) in the hypothalamus, which then sends signals to other regions of the brain and peripheral tissues, including the heart [31]. SCN in the hypothalamus serves as the central circadian pacemaker, orchestrating the synchronization of peripheral clocks throughout the body, including in the heart [32]. Light, detected via melanopsin-containing retinal ganglion cells, transmits signals to the SCN through the retinohypothalamic tract [33]. The SCN processes this light information and generates rhythmic outputs to regulate downstream peripheral clocks. In the cardiovascular system, the SCN entrains the circadian oscillator in cardiac tissues via two primary pathways: hormonal signaling and autonomic nervous system regulation [34,35]. Hormonal signals, such as those mediated by glucocorticoids and melatonin, provide systemic cues that influence clock gene expression in cardiac cells [34,36]. Meanwhile, the autonomic nervous system transmits time-of-day information directly to the heart through sympathetic and parasympathetic innervation, modulating cardiac function and reinforcing local circadian rhythms [37]. These pathways ensure that the central circadian clock aligns the rhythmic activity of peripheral clocks in the heart with environmental light–dark cycles, optimizing cardiac electrophysiology, metabolism, and contractility to meet the demands of daily activity patterns. Disruptions in this hierarchical regulation, such as misalignment between the SCN and peripheral clocks, can lead to desynchronized cardiac rhythms and increase the risk of cardiovascular diseases, including arrhythmias and heart failure. Circadian output pathways are mechanisms by which the circadian clock regulates gene expression and physiological processes in peripheral tissues [38]. In the heart, the circadian clock influences the expression of genes involved in cardiac electrophysiology, metabolism, and inflammation. For example, it regulates the expression of genes involved in calcium handling, such as SERCA2a [39,40,41] and RyR2 [42], which are critical for maintaining proper cardiac rhythm and function. In addition, circadian regulatory factors are proteins that interact with the core oscillator and other components of the circadian clock to fine-tune its regulation, including Bmal1, Clock, Per2, and Rev-erbα. Bmal1, Clock, Per2, and Rev-erbα are both core components of the circadian oscillators involved in the transcription–translation feedback loop and regulators involved in broader gene regulation and physiological function regulation [43,44,45,46] (Figure 1). In the heart, the peripheral circadian oscillator—composed largely of these same clock genes—operates autonomously but remains entrained by SCN-driven signals, ensuring that local cardiac rhythms remain synchronized with the external day–night cycle. Consequently, clock-controlled genes governing ion channel expression, calcium cycling (e.g., SERCA2a, RyR2), and myocardial metabolism are time-regulated, leading to predictable fluctuations in heart rate, blood pressure, and contractility over the 24-h period. This tight coupling between central and peripheral oscillators is crucial for cardiac health; misalignment in these rhythms can exacerbate heart failure, promote arrhythmic events, and contribute to adverse cardiovascular outcomes.

### 2.2. Core Clock Genes and Their Mechanisms

The circadian rhythm in organisms is regulated by a series of core clock genes and clock-controlled genes through a positive and negative feedback loop (Figure 2). The core clock genes include *Clock*, *Bmal1* (also known as *Arntl*), *Cry1/2*, *Per1/2*, *Nr1d1* (Rev-Erb), *Rora*, *Rorb*, and *Rorc* (Roc). Among them, *Bmal1* and *Clock* are positive regulatory genes, while *Cry* and *Per* are negative regulatory genes. Clock-controlled genes mainly consist of D-box binding protein, hepatic leukemia factor, thyrotroph embryonic factor, and Dec1 (*Stra13* or *Sharp2*) [47,48].

Initially, BMAL1 and CLOCK proteins form a heterodimer in the nucleus. This heterodimer binds to regulatory elements (E-box and D-box) on the promoters of *Per* and *Cry* genes, activating their transcription. The protein products then form heterodimers in the cytoplasm, which translocate back to the nucleus to inhibit the transcriptional activity of BMAL1/CLOCK heterodimers on downstream target genes, forming a negative feedback regulatory loop. As the levels of PER/CRY heterodimer proteins decrease and their inhibitory effect on CLOCK/BMAL1 activity wanes, a new transcription–translation cycle begins, thus regulating the 24-h circadian rhythm. Additionally, BMAL1/CLOCK heterodimers also regulate the transcription of orphan nuclear receptor genes *Nr1d1*, *Nr1d2* (Rev-erbα/β) and *Rora*, *Rorb* (Rorα/β) [49]. The translated REV-ERB and ROR proteins compete for binding to the ROR response elements (RRE) in the promoter region of *BMAL1*. This competition binding leads to the activation or inhibition of *BMAL1* transcription depending on which protein binds. ROR proteins activate *BMAL1* transcription, whereas REV-ERB inhibits it, forming another transcriptional feedback regulatory loop for the circadian rhythm [47,48].

### 2.3. Circadian Regulation of Ion Channels and Transporters

Circadian rhythm genes regulate the activity of ion channels on the myocardial cell membrane to regulate cardiac arrhythmias. Specific knockout of the *Bmal1* gene in myocardial cells can lead to loss of tissue repolarization, sinus bradycardia, prolonged QRS duration, and increased frequency of ventricular arrhythmias (VA) [29]. *BMAL1* gene regulates cardiac ion channels, including Nav1.5 [50], Kv1.5 [51], and L-type Ca^2+^ channels [52], influencing myocardial electrophysiology. Original research has demonstrated that cardiac Na^+^ channel (Nav1.5) channels, Kv1.5 channels, and L-type Ca^2+^ channels are regulated by the circadian clock, showing oscillatory behavior in their expression and activity. For example, Schroder et al. demonstrated that Nav1.5 exhibits circadian-related QT interval prolongation, particularly during the nighttime, suggesting that the function and expression of Nav1.5 may be regulated by the circadian rhythm [53]. Yamashita et al. showed that the expression of Kv1.5 in myocardial cells follows a circadian pattern, with peak levels occurring during the dark period (at Zeitgeber time ZT18) and with the lowest levels during the light period (ZT6) [54]. Similarly, Plante et al. reported that L-type Ca^2+^ channel activity varies diurnally, with significant implications for cardiac electrophysiology and susceptibility to arrhythmias [55].

#### 2.3.1. Circadian Rhythm and Myocardial Depolarization

Nav1.5 are widely expressed in the sinus node, Purkinje fibers, and myocardial cells of mammals, playing a key role in mediating myocardial depolarization [56]. Complete knockout of the *SCN5A* gene in mice leads to abnormal ventricular development and structural defects, resulting in intrauterine death [57]. Nav1.5 +/− heterozygous mice can be delivered normally, but still exhibit deficiencies in atrioventricular conduction tissue, delayed myocardial conduction, and VT [58]. Hence, the downregulation of Nav1.5 expression is a crucial mechanism contributing to the prolonged depolarization of myocardial cells and is essential in the processes of ventricular arrhythmias (VAs).

Studies investigating the circadian regulation of Nav1.5 have shown that its expression exhibits significant circadian rhythmic changes. For example, Nav1.5, encoded by the *SCN5A* gene, exhibits circadian rhythm-dependent regulation, and disruptions in environmental time cues, such as light and feeding schedules, can exacerbate the LQT3-related phenotype by affecting the circadian rhythm of QT intervals [53]. When compared to wild-type mice, cardiomyocyte-specific knockout of the *BMAL1* gene in adult mice showed reduced Nav1.5 expression, inward currents, and heart rate; prolonged RR intervals and QRS durations; and increased susceptibility to electrically induced arrhythmias [59]. The research also demonstrated that sodium was found to have a circadian pattern of expression in control hearts but not in the hearts of mice with *Bmal1* deletion, and the loss of *SCN5A* circadian expression in these knockout hearts was associated with decreased levels of Nav1.5 and a reduction in sodium current (I_na_) in ventricular myocytes [59]. Therefore, the evidence suggests that circadian rhythms regulate Nav1.5, which may influence myocardial depolarization processes and thereby either counteract or contribute to the occurrence of ventricular arrhythmias. Moreover, variants in *SCN5A* are associated with a broad range of primary arrhythmic characteristics [60]. Most variants linked to dilated cardiomyopathy (DCM) lead to a multifocal, ventricular premature beat (VPB)-dominant form of cardiomyopathy that can be managed with sodium channel-blocking medications.

#### 2.3.2. Circadian Rhythm and Myocardial Repolarization

The prolongation of the ventricular AP is a key mechanism in the occurrence and maintenance of VA, with prolonged repolarization of ventricular muscle potentially leading to VF or SCD [61]. Evidence indicates that Krüppel-like factor 15 (KLF15) regulates cardiac repolarization by modulating the interaction of K^+^ channel auxiliary subunit K^+^ channel interacting protein 2 (KChIP2), which is responsible for the transient outward K^+^ current responsible for repolarization in myocardial cells [62]. Interestingly, the expression of KLF15 is regulated by CLOCK/BMAL1 heterodimers, suggesting that circadian rhythm genes may influence cardiac repolarization [51].

Mice exposed to a dark environment for 36 h showed significant circadian rhythm in the expression of the α subunit (Kv4.2) of the transient outward K^+^ current and the regulatory β subunit (KChIP2). Knockout of KLF15 resulted in a prolonged QT interval in ventricular myocardium, while overexpression of KLF15 led to a shortened QT interval and changes in the ST segment [51]. These findings are consistent with the effects observed from knockout or overexpression of *BMAL1*, indicating that circadian rhythm genes regulate myocardial cell repolarization by controlling the transcription and protein expression of KLF15 [63]. Similarly, Schröder et al. [64] found that *KCNH2*, as a direct target of BMAL1 in myocardial cells, was responsible for regulating myocardial repolarization. Under normal lighting conditions, both *BMAL1* knockout mice and *KCNH2* knockout mice exhibit significantly prolonged QT intervals.

These studies demonstrate that changes in the expression of circadian rhythm genes *CLOCK/BMAL1* can affect myocardial repolarization by regulating K^+^ channel-related proteins in myocardial cells, thereby influencing the development and persistence of VA and SCD.

#### 2.3.3. Impact on Electromechanical Activity in Myocardial Cell

Luciferase reporter gene results suggest that overexpression of BMAL1 can regulate the *SCN5A* mRNA level [59]. In addition, Schroder et al. demonstrated that potassium channel-related regulatory genes in myocardial cells exhibit significant circadian rhythm changes [65]. The regulation of ion channels, including Kv1.5 (encoded by *KCNA5*), by circadian rhythm genes has been a subject of interest due to its potential impact on cardiac electrophysiology. However, it is essential to clarify that, for a behavior or gene expression pattern to be classified as circadian, it must exhibit oscillations that persist independently of environmental cues, such as constant darkness. From 6 a.m. to 6 p.m., mRNA levels of the voltage-gated potassium channel (Kv1.5) in myocardial cells approximately double. Blocking light stimulation nearly abolishes these circadian rhythm changes in the K^+^ channel, while autonomic nerve blockade only partially weakens the circadian rhythm of Kv1.5 [54]. This study demonstrated that the oscillatory behavior of *KCNA5* expression ceased under constant dark conditions, indicating that its rhythmic expression is not truly circadian but rather dependent on external environmental factors. This finding challenges the initial assumption that Kv1.5 is regulated by an intrinsic circadian clock [54]. Changes in circadian rhythm can affect the activity of multiple ion channels in myocardial cells. Both sustained overexpression or knockout of circadian rhythm genes can influence the electromechanical activity of myocardial cells and induce cardiac arrhythmias [65]. The overexpression of BMAL1 has been shown to alter the expression of several ion channel genes, including *KCNA5* (encoding KV1.5) [51]. However, these changes in gene expression do not necessarily indicate that these genes are oscillatory in a physiologically significant manner. The gain-of-function experiments primarily demonstrate BMAL1’s capacity to influence gene expression, but further studies are needed to determine whether or not these effects have meaningful implications under normal physiological conditions.

## 3. Effects of Circadian Clock Proteins in Modulating Myocardial Electrophysiology

### 3.1. Circadian Regulation of Ion Channels

The mouse model of heart failure ventricular AP consists of a rapid depolarization phase, an instantaneous repolarization phase, and a slow repolarization phase (with no apparent plateau phase in mouse ventricular AP) [66]. Fast depolarization is caused by sodium channel current (I_na_) flowing through the Nav1.5 channel encoded by *SCN5A* [67]. The instantaneous and slow ventricular repolarization is induced by various potassium channel currents (I_to,_ f, I_to,_ s, I_K_slow_1_, I_K_slow_2_, I_SS_, and I_Ks_) and calcium channel current (I_a,_ L).

#### 3.1.1. Sodium Channel

*SCN5A* expression in the mouse ventricle exhibits circadian rhythmic changes, oscillating with a 24-h period in constant darkness. In the hearts of *BMAL1* knockout (*BMAL1*^−/−^) mice, the circadian rhythmic oscillation of *SCN5A* transcription levels is lost, and in isolated voltage-clamped ventricular myocytes, the peak I_na_ is reduced by 30% compared to the control group [68].

#### 3.1.2. Potassium Channels

*KCND2* and *KCNH2* mRNA oscillate with a 24-h period in constant darkness, suggesting circadian rhythmic oscillation in mouse ventricles at the transcriptional level [64]. In the hearts of *BMAL1*^−/−^ mice, the circadian rhythmic oscillation of *KCND2* transcription levels is lost, and in isolated voltage-clamped ventricular myocytes, the peak I_kr_ is reduced by 50% compared to the control group [59]. Moreover, *Bmal1* directly regulates the circadian expression of KLF15. In earlier studies, using transgenic mice, it was shown that KLF15 drives the circadian expression of *KCNI2* [69,70]. Furthermore, compared to the control group, *BMAL1*^−/−^ hearts showed reduced transcription levels of *KCNIP2*; most ventricular myocytes isolated from *BMAL1*^−/−^ hearts lacked I_to,_ f, and *BMAL1*^−/−^ ventricular AP duration was longer than control myocardial cells [71]. Consistent with this data, decreased transcription levels of critical potassium channels, such as *KCNIP2* and *KCNA5*, were found in *BMAL1*^−/−^ hearts [72]. Hayter et al. [73] found that the expression of *Kcne1* increased in αMHCCREBmaL1fl/fl mice, and *Kcne1* overexpression increased susceptibility to atrial fibrillation. Taken together, the expression changes in clock genes affect myocardial repolarization by regulating the expression of potassium channel genes in myocardial cells, thereby participating in the occurrence and maintenance of arrhythmias.

#### 3.1.3. Calcium Channels

Chen et al. [52], using patch-clamp techniques, detected that the activity of L-type calcium channels in myocardial cells peaked at 3 a.m., and CLOCK/BMAL1 overexpression significantly inhibited the α-subunit expression of L-type calcium channels, reducing the current levels. In addition, CLOCK/BMAL1 overexpression significantly decreased the level of *Cacna1c*, and αMHCCREBmaL1fl/fl mice *Cacna1c* expression weakened, while *Cacna1c* gene mutation disrupted the regulation of Cav1.2 on voltage and Ca^2+^ in cardiac myocytes, causing abnormal calcium current, affecting myocardial cell AP and triggering arrhythmias [74]. Calsequestrin (CASQ) regulates sarcoplasmic reticulum Ca^2+^ release and heart rate in the heart. Lack of CASQ1 causes malignant hyperthermia/environmental heat-stroke-like ventricular arrhythmias [75], while αMHCCREBmaL1fl/fl mouse CASQ1 and CASQ2 expression significantly increased [73], suggesting that the expression changes in CLOCK/BMAL1 circadian genes affect ventricular repolarization by regulating the expression of calcium channel genes in myocardial cells, leading to arrhythmias. The details are shown in Table 1, and a summary of the animal models mentioned in the review is shown in Table 2.

In addition, Table 3 was created to summarize the impact of circadian mechanisms on different ion channels.

## 4. Clinical Implications and Therapeutic Potential Approaches

### 4.1. Overview of Current Treatments for Heart Failure-Related Arrhythmias

Treatment strategies for heart failure-related arrhythmias focus on managing symptoms, improving cardiac function, and reducing the risk of life-threatening arrhythmic events. This approach typically involves a combination of pharmacological, device-based, and in some cases, procedural interventions.

#### 4.1.1. Pharmacological Therapy

The current guidelines recommend certain medications as first-line therapy for heart failure, such as beta-blockers, angiotensin-converting enzyme (ACE) inhibitors or angiotensin II receptor blockers (ARBs), mineralocorticoid receptor antagonist (MRA), and sodium-glucose cotransporter 2 inhibitors (SGLT2i) [76]. Beta-blockers such as carvedilol [77], metoprolol [78], and bisoprolol [79] are commonly prescribed. It is recommended for patients with systolic heart failure and a HR above 70 bpm, either despite treatment with beta-blockers or in cases of beta-blocker intolerance [80]. Furthermore, antiarrhythmic medications like amiodarone [81], dofetilide [82,83], and sotalol [84] may be used to control specific types of arrhythmias, particularly atrial fibrillation, or VT. However, their use requires careful monitoring due to potential side effects. ACE inhibitors (e.g., ramipril) and ARBs (e.g., valsartan) reduce the workload on the heart and may help prevent arrhythmias by improving cardiac function. ARNIs (angiotensin receptor-neprilysin inhibitors, such as sacubitril/valsartan) are used for patients who cannot tolerate ACE inhibitors. Sacubitril/valsartan, a drug belonging to ARNIs (angiotensin receptor-neprilysin inhibitors), may possess antiarrhythmic properties and reduce the risk of ventricular arrhythmias and sudden cardiac death in patients with heart failure with reduced ejection fraction (HFrEF) [85]. MRA, such as spironolactone and eplerenone, work by blocking aldosterone receptors, thereby reducing sodium and water retention. This action alleviates the burden on the heart and has been shown to decrease mortality and hospitalization rates in patients with heart failure [76,86,87,88]. SGLT2i, such as empagliflozin and dapagliflozin, have emerged as a cornerstone in the treatment of HFrEF [89]. According to the 2022 European Society of Cardiology (ESC) guidelines, these agents are recommended as first-line therapy for patients with HFrEF, regardless of the presence of diabetes [90]. Specifically, by inhibiting the myocardial sodium–hydrogen exchanger (Na^+/^H^+^ exchanger), which is upregulated in heart failure, SGLT2i can improve mitochondrial dysfunction and reduce oxidative stress, thereby lowering the risk of arrhythmias [91,92]. Major trials like PRESERVED-HF and EMPEROR-Reduced have demonstrated these benefits, further solidifying their role in clinical practice [93,94]. Furthermore, compared to digoxin (Inotropes) alone, combination therapy with carvedilol and digoxin in patients with atrial fibrillation and heart failure resulted in a lower ventricular rate on 24-h ambulatory electrocardiographic monitoring and during submaximal exercise, while also improving left ventricular ejection fraction and symptom scores [95]. Additionally, diuretics like furosemide are used to manage fluid retention, reducing the strain on the heart and minimizing the risk of arrhythmias associated with volume overload. Recent evidence by Mariani MV et al. [96] demonstrated significant antiarrhythmic benefits of SGLT2i in patients with HF with HFrEF. In this study, involving 198 HFrEF patients with implantable cardioverter defibrillators or cardiac resynchronization therapy defibrillators (CRT-Ds), a notable reduction (73.8%) in both atrial and ventricular arrhythmias (AAs and VAs) was observed one year after initiating SGLT2i therapy. The median number of arrhythmic episodes decreased significantly, including episodes of atrial fibrillation, non-sustained ventricular tachycardia (NSVT), and sustained ventricular tachycardia (SVT). These findings provide compelling clinical evidence supporting the molecular insights discussed in our manuscript. They clearly illustrate how modulating ion channels and circadian regulatory mechanisms—pathways influenced by SGLT2i—can translate into improved clinical outcomes for HF patients, specifically through substantial antiarrhythmic effects.

#### 4.1.2. Device-Based Therapy

Device-based therapy includes Implantable Cardioverter-Defibrillators (ICDs) and Cardiac Resynchronization Therapy (CRT). Implantable Cardioverter-Defibrillators (ICDs) are critical devices in the management of heart failure patients at risk of sudden cardiac death due to life-threatening arrhythmias [97]. ICDs continuously monitor the heart’s rhythm and can deliver immediate therapeutic interventions, such as anti-tachycardia pacing (ATP) or defibrillation shocks, when they detect potentially fatal arrhythmias like VT or VF [98,99]. By providing these life-saving interventions, ICDs play a crucial role in reducing the risk of sudden cardiac death in high-risk patients. In patients with heart failure, early intervention with CRT-D reduces mortality, ventricular tachyarrhythmia (VTA) burden, and the frequency of multiple appropriate implantable ICD shocks [100].

#### 4.1.3. Procedural Interventions

Procedural interventions include ablation and cardiac surgery. Catheter ablation may be considered for certain arrhythmias, especially atrial fibrillation, or ventricular tachycardia. This procedure involves selectively destroying or isolating abnormal electrical pathways in the heart. In some cases, particularly when structural heart disease is present, surgical interventions like the Maze procedure may be performed to eliminate or control atrial fibrillation.

#### 4.1.4. Lifestyle Modifications

Adopting a heart-healthy diet low in sodium and saturated fats, along with regular exercise, can improve overall heart health and help manage arrhythmias. Additionally, quitting smoking and moderating alcohol intake can positively impact heart health and reduce the risk of arrhythmias.

#### 4.1.5. Education and Monitoring

Education and monitoring are also critical components in the management of heart failure. Providing patients with comprehensive information about their condition, medications, and the importance of adherence to treatment plans is essential for effective management. In addition, close monitoring of cardiac function through regular check-ups, electrocardiograms (ECGs), and other diagnostic tests enables healthcare providers to adjust treatment plans as needed. The choice of treatment approach depends on the type and severity of the arrhythmia, the underlying cause of heart failure, and individual patient characteristics. A tailored, multidisciplinary care involving collaboration between cardiologists, electrophysiologists, and other healthcare professionals is essential for optimal outcomes in managing heart failure-related arrhythmias.

### 4.2. Potential Interventions Targeting Circadian Rhythm Proteins and Epigenetic Pathways

Epigenetic modifications enable organisms to adapt to stress and environmental fluctuations, allowing dynamic responses to both external and internal stimuli. However, when dysregulated, these modifications can contribute to pathological conditions, such as heart failure [101]. Emerging research highlights the complex interplay between circadian rhythms, epigenetic regulation, and the development of heart failure-associated arrhythmias. A study reported by Ferreira et al. [102] found that doxorubicin (DOX) treatment leads to significant and lasting disruptions in the circadian regulation of the heart, including alterations in the expression of circadian genes and the acetylation of proteins like BMAL1, which are involved in circadian rhythms. These disruptions are linked to DOX-induced cardiotoxicity and could contribute to long-term cardiovascular outcomes. The document does connect circadian clock proteins with epigenetic pathways (specifically protein acetylation) in the context of DOX-induced heart damage. Furthermore, specific nuclear receptors involved in circadian rhythm regulation, such as RORα (Retinoic Acid Receptor-Related Orphan Receptor Alpha), play a crucial role in cardiac function. Research has shown that the deficiency of RORα exacerbates cardiac diseases, including cardiac hypertrophy and dysfunction [103]. Furthermore, Berulava et al. [104] revealed that the m6A landscape is altered in heart hypertrophy and heart failure, with changes in m6A RNA methylation leading to alterations in protein abundance independent of mRNA levels, suggesting that targeting epi-transcriptomic processes, such as m6A methylation, could be a promising approach for therapeutic interventions on heart failure. Additionally, Koczor et al. [105] demonstrated that MDMA causes differential expression of circadian rhythm genes such as *Per3*, CLOCK, *ARNTL*, and *NPAS2*, and these changes are associated with hypermethylation of cardiac DNA. Importantly, these gene expression alterations persist beyond MDMA exposure, suggesting that circadian–epigenetic interactions may have lasting impacts on cardiac pathophysiology.

Given the intricate connections between circadian rhythms, epigenetic modifications, and cardiac pathophysiology, several promising therapeutic strategies have emerged. Chronotherapy, which aligns pharmacological treatments with circadian rhythms, offers the potential to optimize therapeutic efficacy and minimize adverse effects, particularly in heart failure patients experiencing circadian disruptions. Additionally, pharmacological modulation of epigenetic enzymes—such as histone acetyltransferases/deacetylases and DNA methyltransferases—could help restore normal circadian gene expression patterns and mitigate heart failure progression. Novel epi-transcriptomic therapies targeting RNA modifications, particularly m6A RNA methylation, represent another promising approach by stabilizing protein expression independently of transcriptional alterations. Finally, lifestyle interventions and environmental modifications, including adjusted light exposure, dietary timing, and improved sleep patterns, may effectively reset disrupted circadian rhythms and provide non-pharmacological means to alleviate cardiac dysfunction associated with circadian disturbances. Overall, this study highlights the crucial role of circadian clock proteins and epigenetic pathways in heart failure-related arrhythmias, particularly through the modulation of gene expression via DNA methylation, which may underlie circadian rhythm disturbances in the heart [105].

### 4.3. Chrono-Pharmacology Represents a Promising Therapeutic Approach

Chrono-pharmacology represents a promising therapeutic approach that leverages the intrinsic circadian rhythms of biological processes to optimize drug efficacy and reduce side effects. By timing medication administration to coincide with specific circadian phases of maximal therapeutic responsiveness or minimal toxicity, chrono-pharmacology can enhance clinical outcomes in patients with heart failure. Recent studies have demonstrated that drugs commonly prescribed for cardiovascular conditions, including antihypertensive agents [106], anticoagulants [107], and antiarrhythmics [108], exhibit significant circadian variation in their pharmacokinetics and pharmacodynamics [109]. Thus, strategically aligning medication timing with circadian peaks and troughs of molecular targets or physiological pathways could improve drug efficacy, reduce adverse events, and potentially enhance patient compliance. Further in-depth exploration and clinical trials of chrono-pharmacological strategies are warranted to fully harness their potential benefits and to develop precise, personalized treatment regimens for HF and associated arrhythmias.

## 5. Conclusions

This article reviews the mechanisms by which core clock genes regulate myocardial electrical remodeling in the context of heart failure-related cardiac arrhythmias through epigenetic pathways. This regulation involves the modulation of myocardial cell membrane Na^+^, K^+^, and Ca^2+^ channels, impacting the mechanisms of myocardial cell depolarization and repolarization.

However, most studies on arrhythmias have been conducted in mice, whose cardiac structure is less complex than that of humans and involves fewer genes and ion channels. Consequently, further research involving human studies is essential. Understanding the mechanisms of the circadian clock, especially through epigenetic pathways, is crucial for developing new therapeutic drugs. Administering medication at optimal times could significantly reduce cardiovascular risks for shift workers and individuals with cardiovascular diseases.

## Figures and Tables

**Figure 1 ijms-26-02728-f001:**
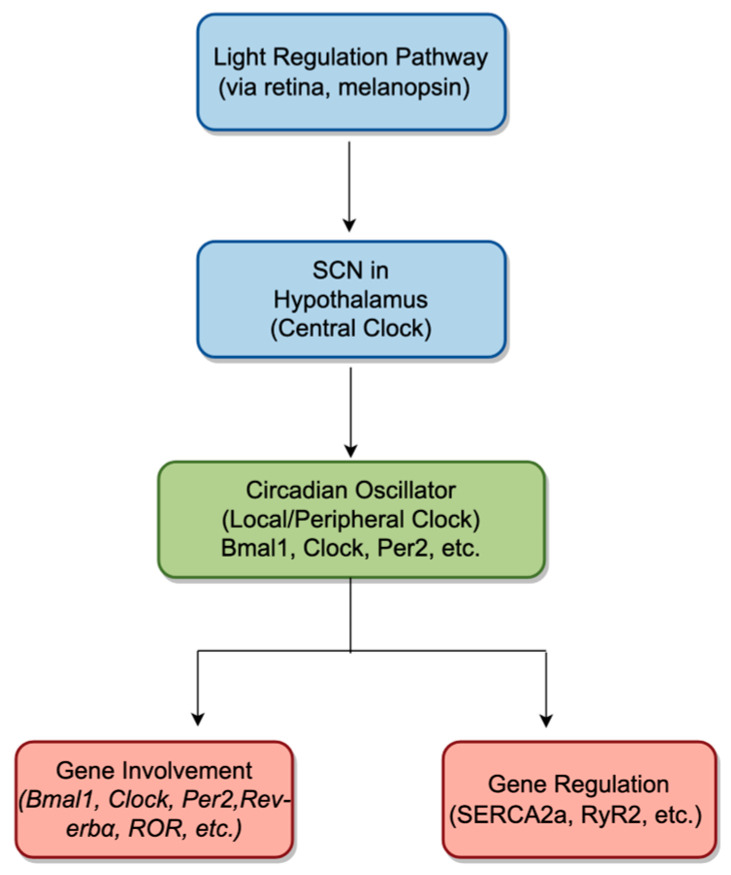
Schematic representation of central and local circadian regulatory mechanisms and their influence on cardiac function. Light signals, detected by melanopsin-expressing retinal cells, travel to the suprachiasmatic nucleus (SCN) in the hypothalamus (the central clock). The SCN then entrains the local (peripheral) circadian oscillator in the heart—depicted here by the green box—via hormonal signals and autonomic nervous system pathways. This cardiac oscillator comprises key clock genes (e.g., *Bmal1*, *Clock*, *Per2*) and in turn regulates downstream gene expression relevant to calcium handling (e.g., SERCA2a, RyR2) and other cardiac functions. The interplay between these central and local clocks ensures synchronization of the cardiovascular system with the external light–dark cycle. The blue boxes and arrows illustrate the primary circadian pathways and flow of information, while the red boxes indicate genes and regulatory targets central to these processes.

**Figure 2 ijms-26-02728-f002:**
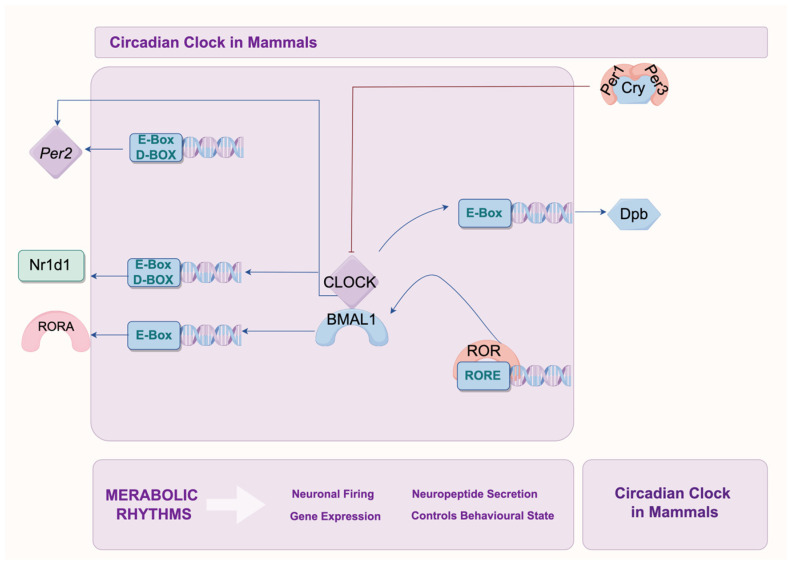
The circadian rhythm in humans is regulated via core clock genes and clock-controlled genes within the positive and negative feedback loops. This figure illustrates how BMAL1 and CLOCK proteins form a heterodimer in the nucleus and bind to regulatory elements on the promoters of *Per* and *Cry* genes, activating their transcription. The protein products then form heterodimers in the cytoplasm and translocate to the nucleus to inhibit the transcriptional activity of BMAL1/CLOCK heterodimers on downstream target genes, forming a negative feedback regulatory loop. Additionally, BMAL1/CLOCK heterodimers activate the transcription of orphan nuclear receptor genes *Nr1d1*, *Nr1d2* (Rev-erbα/β)and *Rora*, *Rorb* (RORα/β). These proteins compete for binding to ROR response elements in the *BMAL1* promoter region, either activating or inhibiting *BMAL1* transcription, which forms another transcriptional feedback regulatory loop governing circadian rhythm.

**Table 1 ijms-26-02728-t001:** Circadian genes alter the expression of cardiac channel proteins.

Channel Type	Genes	Expression Change in BMAL1^−/−^ Hearts	Functional Impact in Myocardial Cells	Reference
Sodium Channel	*SCN5A*	Decreased expression	Reduction of peak I_na_ by 30% in ventricular myocytes	[61]
Potassium Channel	*KCND2*, *KCNH2*	Loss of expression	Reduction of I_kr_ peak by 50% in ventricular myocytes	[30]
	*KLF15*, *KCNIP2*	Loss of circadian expression	Reduced transcription levels of Kcni2; lack of I_to,_ f	[62]
	*KCNIP2*, *KCNA5*	Decreased transcription levels	Unknown impact on ventricular myocytes	[51]
	*KCNE1*	Increased expression	Increased susceptibility to atrial fibrillation	[63]
Calcium Channel	*CACNA1C*	Weakened expression	Disrupted regulation on voltage and Ca^2+^ in cardiac myocytes	[64]
	*CASQ1*, *CASQ2*	Increased expression	Possible regulation on sarcoplasmic reticulum Ca^2+^ release and heart rate	[59,63]

Note: *BMAL1*^−/−^: Knockout of BMAL1 (Brain and Muscle ARNT-Like 1), a core component of the circadian clock. I_na_: Sodium current, critical for cardiac depolarization. I_kr_: Rapid delayed rectifier potassium current, involved in cardiac repolarization. I_to,_ f: Transient outward potassium current, important for early repolarization. Ca^2^⁺: Cal cium ion, essential for cardiac excitation–contraction coupling. *SCN5A*: Sodium Voltage-Gated Channel Alpha Subunit 5; *KCND2*: Potassium Voltage-Gated Channel Subfamily D Member 2; *KCNH2*: Potassium Voltage-Gated Channel Subfamily H Member 2; *KLF15*: Krüppel-Like Factor 15; *KCNIP2***:** Potassium Voltage-Gated Channel; Modifier Subfamily I Member 2; *KCNA5*: Potassium Voltage-Gated Channel Subfamily A Member 5; *KCNE1*: Po tassium Voltage-Gated Channel Subfamily E Regulatory Subunit 1; *CACNA1C*: Calcium Voltage-Gated Channel Alpha Subunit 1C; *CASQ1*, *CASQ2*: Calsequestrin 1 and 2.

**Table 2 ijms-26-02728-t002:** Summary of key animal studies.

Study	Animal Model	Major Findings	Implications
BMAL1 Knockout Studies	Mouse (*Bmal1*^−/−^)	Loss of circadian rhythmic oscillation in sodium (*SCN5A*) and potassium channels (*KCND2, KCNH2*), leading to impaired repolarization and increased arrhythmia susceptibility.	BMAL1 is crucial for maintaining circadian control of ion channels, impacting myocardial electrophysiology and arrhythmia risk.
Nav1.5 Channel Oscillation	Mouse	Circadian rhythmic oscillation in Nav1.5 expression affects myocardial depolarization; disrupted in *Bmal1*^−/−^ models.	A direct role of BMAL1 in regulating sodium channel activity and the cardiac action potential.
Clock Gene Impact on Potassium Channels	Transgenic mice	BMAL1/CLOCK overexpression reduces potassium channel expression, affecting repolarization.	The role of circadian regulation in maintaining cardiac electrical stability.
L-Type Calcium Channel Regulation	Patch-clamp study in mouse myocardial cells	Clock/BMAL1 overexpression reduces L-type calcium current, leading to arrhythmia.	Demonstrates how circadian disruptions in calcium handling contribute to arrhythmic risk.
Epigenetic Impact of Doxorubicin (DOX)	Mouse model	DOX disrupts circadian homeostasis and alters protein acetylation, increasing cardiotoxicity.	Epigenetic modulation of circadian genes may offer therapeutic insights into reducing DOX-induced cardiotoxicity.
RORα Deficiency	Mouse (*Rora*^−/−^)	Exacerbates cardiac hypertrophy and dysfunction by impairing mitochondrial function.	RORα is critical for circadian regulation of mitochondrial biogenesis and cardiac function, with implications for heart failure therapy.
Histone Modification Effects	Mouse models	Changes in histone methylation (e.g., H3K4) impact ion channel expression and arrhythmogenesis.	Epigenetic mechanisms, like histone methylation, influence cardiac repolarization and arrhythmic susceptibility.

Note: *Bmal1*^−/−^: Brain and Muscle ARNT-Like 1 Knockout; *SCN5A*: Sodium Voltage-Gated Channel Alpha Subunit 5; *KCNH2*: Potassium Voltage-Gated Channel Subfamily H Member 2; Nav1.5: Voltage-Gated Sodium Channel Nav1.5; *Rora*^−/−^: Retinoic Acid Receptor-Related Orphan Receptor Alpha Knockout; H3K4: Histone 3 Lysine 4 Methylation; *Clock*/BMAL1: Circadian locomotor output cycles kaput/Brain and muscle Arnt-like protein 1.

**Table 3 ijms-26-02728-t003:** Impact of Circadian Mechanisms on Ion Channels in Cardiac Myocytes.

Ion Channel Type	Gene	Circadian Regulation	Effects of Circadian Disruption	Functional Consequence
Sodium (Na⁺) Channel	*SCN5A*	24-h rhythmic oscillation in expression	Loss of circadian rhythmic expression; reduction in peak I_na_ by 30%	Impaired depolarization, increased arrhythmia susceptibility
Potassium (K⁺) Channel	*KCND2*, *KCNH2*	24-h rhythmic oscillation in expression	Loss of circadian rhythmic expression; reduction in peak I_kr_ by 50%	Prolonged action potential duration, impaired repolarization
	*KLF15*, *KCNIP2*	Circadian regulation by KLF15	Reduced transcription levels; loss of I_to,_ f, prolonged ventricular AP duration	Impaired repolarization, prolonged AP duration
	*KCNIP2*, *KCNA5*	Circadian expression regulated by BMAL1	Decreased transcription levels	Potential impaired repolarization
	*KCNE1*	Circadian regulation	Increased expression in BMAL1-altered mice; increased AF susceptibility	Increased susceptibility to atrial fibrillation
Calcium (Ca^2^⁺) Channel	*CACNA1C*	Circadian rhythmic oscillation; regulated by CLOCK/BMAL1	Weakened expression; reduced L-type calcium current	Abnormal calcium currents, triggering arrhythmias
	*CASQ1*, *CASQ2*	Circadian gene regulation by CLOCK/BMAL1	Increased expression in BMAL1-altered mice	Dysregulated sarcoplasmic reticulum Ca^2^⁺ release, arrhythmogenic potential

Note: AP: Action potential; AF: Atrial fibrillation; I_na_: Sodium current; I_kr_: Rapid delayed rectifier potassium current; I_to,_ f: Transient outward potassium current, fast component.

## Data Availability

No new data were created or analyzed in this study.

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
