# Peer review of "Heart Failure and Arrhythmias: Circadian and Epigenetic Interplay in Myocardial Electrophysiology"

_ijms, 2025, doi:10.3390/ijms26062728_

Round 1
Reviewer 1 Report (New Reviewer)
Comments and Suggestions for Authors
Congratulations to the author for having produced a manuscript about such an interesting and promising field of therapeutic targets such as the influence of circadian cycle on HF and arrhythmias; here my comments on the paper:
You might consider including a summary figure or table that illustrates the impact of circadian mechanisms on different ion channel types (Na+, K+, Ca2+), providing a clearer comparison of their regulation.
The role of BMAL1 and CLOCK in regulating ion channels is mentioned repeatedly throughout the manuscript, resulting in redundant content that could be streamlined for clarity.
The manuscript should more clearly establish how molecular discoveries translate into practical treatment strategies for heart failure. Regarding this authors should briefly include and discuss the latest evidencies regarding the pleiotropic effects of HF drugs which prove the interconnection between HF and arrhythmias (doi: 10.1002/ehf2.15223.)
Additionally, the manuscript would benefit from a more detailed discussion of potential therapeutic interventions related to the described circadian and epigenetic mechanisms.
Furthermore, promising therapeutic strategies like chrono-pharmacology, which optimizes drug administration timing based on circadian rhythms, should be explored in greater depth.
Author Response
Comments 1: You might consider including a summary figure or table that illustrates the impact of circadian mechanisms on different ion channel types (Na+, K+, Ca2+), providing a clearer comparison of their regulation.
Response 1: We sincerely thank the reviewer for their valuable suggestion to include a summary table clearly illustrating the impact of circadian mechanisms on different ion channels in Table 3 on Page 10-11. As recommended, we have created a summary table that clearly compares the circadian regulation, the effects of circadian disruption (such as in BMAL1 knockout or CLOCK/BMAL1 overexpression models), and the resulting functional consequences for sodium (Na⁺), potassium (K⁺), and calcium (Ca²⁺) channels in myocardial cells. We believe this table effectively highlights the critical points discussed and improves the clarity and comprehensibility of our manuscript.
Comments 2: The role of BMAL1 and CLOCK in regulating ion channels is mentioned repeatedly throughout the manuscript, resulting in redundant content that could be streamlined for clarity.
Response 2: We appreciate the reviewer’s insightful observation regarding the redundancy of BMAL1 and CLOCK roles throughout the manuscript. We have streamlined the text to eliminate repetitive statements and enhance clarity. As suggested, we have consolidated the discussion on circadian regulation by BMAL1 and CLOCK into concise, comprehensive statements. A comparison table is provided above to clearly show the original repetitive sentences alongside the revised streamlined versions. We believe these revisions significantly improve the manuscript’s readability and focus.
Comments 3: The manuscript should more clearly establish how molecular discoveries translate into practical treatment strategies for heart failure. Regarding this authors should briefly include and discuss the latest evidencies regarding the pleiotropic effects of HF drugs which prove the interconnection between HF and arrhythmias (doi: 10.1002/ehf2.15223.)
Response 3: We appreciate the reviewer’s valuable suggestion to clarify how molecular findings translate into practical treatment strategies for heart failure. Following your recommendation, we have briefly incorporated recent evidence highlighting the pleiotropic effects of sodium-glucose cotransporter 2 inhibitors (SGLT2i), which establish an interconnection between HF and arrhythmias. The added text is on Page 12, Line 449-460.
Comments 4: Additionally, the manuscript would benefit from a more detailed discussion of potential therapeutic interventions related to the described circadian and epigenetic mechanisms.
Response 4: Thank you for your suggestion about our paper. I have revised your manuscript section (Page 13, Line 521-534) to include a detailed discussion on potential therapeutic interventions related to circadian and epigenetic mechanisms, addressing your suggestion.
Comments 5: Furthermore, promising therapeutic strategies like chrono-pharmacology, which optimizes drug administration timing based on circadian rhythms, should be explored in greater depth.
Response 5: We appreciate the reviewer's valuable suggestion regarding chrono-pharmacology as a promising therapeutic strategy. In response, we have expanded our discussion to highlight the importance and potential benefits of chrono-pharmacology in optimizing drug efficacy and reducing adverse effects through precise alignment of drug administration timing with the body's inherent circadian rhythms. Indeed, recent evidence supports chrono-pharmacological approaches as effective in improving clinical outcomes in heart failure patients by ensuring drug delivery coincides with periods of maximum therapeutic responsiveness while minimizing off-target effects. We have now explicitly integrated this concept into our revised manuscript to emphasize its relevance and encourage further research in this promising area. The suppled content is in Page 13-14, Line 538-553.
Reviewer 2 Report (New Reviewer)
Comments and Suggestions for Authors
Scientists have recently discovered a ‘circadian clock’ in the heart that plays a crucial role in regulating
numerous cardiovascular functions, such as endothelial activity, thrombus formation, blood pressure and
heart rate. This circadian system consists of a network of proteins that exhibit varying levels of expression
and activity throughout the day, thus regulating various physiological processes in the body. Circadian clock
proteins are closely linked to the electrophysiological remodelling of the myocardium and epigenetic
pathways associated with arrhythmias in heart failure, leading to electrophysiological remodelling of
cardiomyocytes, which can precipitate life-threatening arrhythmias such as ventricular tachycardia (VT) and
ventricular fibrillation (VF). The aim of this work is to provide insights into potential therapeutic targets and
interventions that may counteract the negative effects of circadian rhythm dysregulation on cardiovascular
health by analysing the connections between circadian rhythms, myocardial electrophysiology and heart
failure-related arrhythmias.
I have the following concerns:
Introduction
Lines 53-68
“Ventricular arrhythmias (VAs) can be categorized into premature ventricular contractions (PVCs),
ventricular tachycardia (VT), ventricular flutter (VF), and ventricular fibrillation (VF), with PVCs and
sustained VT being the most clinically prevalent. Chronic heart failure combined with PVCs can trigger life-
threatening arrhythmias. VT can be categorized based on the patient's hemodynamic status into
hemodynamically stable and unstable VT. Stable VT is often asymptomatic or associated with mild
symptoms, while unstable VT can present with severe symptoms such as syncope, blackout, cardiac arrest,
or even sudden cardiac death. VT episodes are further be classified based on their duration into non-
sustained (duration <30 seconds) and sustained VT (duration >30 seconds or, if <30 seconds, requires urgent
termination due to severe hemodynamic compromise). In chronic heart failure patients, reentrant VT may
occur, particularly involving the His bundle (or its distal portion), the His-Purkinje System, and the
ventricular myocardium. Additionally, bundle branch reentrant VT may be observed, particularly in patients
with dilated cardiomyopathy. The fundamental pathogenesis of arrhythmia can be attributed to
abnormalities in automaticity, reentry, and depolarization activity (early or delayed abnormal
depolarization).”
I suggest streamlining the introduction by eliminating this very specific and descriptive part on ventricular
arrhythmias and moving it, perhaps, to the following paragraphs. In this way, the focus of the discourse is
not lost.
Lines 80-84
“Jeffrey C. Hall, Michael Rosbash, and Michael W. Young were awarded the Nobel Prize in Physiology or
Medicine in 2017 for their groundbreaking discoveries of the molecular mechanisms governing circadian
rhythms. They successfully identified the genes that regulate circadian processes and elucidated how
various components within this mechanism interact to maintain biological rhythms (18).”
I suggest removing this part from the introduction because it is superfluous and unnecessary.
Lines 88-92
“Disruption of these 24-hour rhythms contributes to the onset of cardiovascular diseases, including heart
failure, myocardial infarction, and arrhythmias. Additionally, 24-hour rhythms influence the development,
risk factors, incidence, and outcomes associated with cardiovascular diseases. Cardiovascular disease, in
turn, disrupts circadian rhythms and contributes to sleep disturbances.”
I suggest changing the period with: “Alteration of these 24-hour rhythms contributes to the onset of
cardiovascular diseases, including heart failure, myocardial infarction and arrhythmias, by altering the
normal circadian rhythm and contributing to sleep disturbances. Moreover, this alteration affects
the development, risk factors, incidence and outcomes associated with cardiovascular disease.”
I suggest deleting this period, because it is a repetition of what has already been said in the previous lines.
Lines 110-114
“By exploring the connections between circadian rhythms, myocardial electrophysiology, and heart failure-
related arrhythmias, this review aims to provide insights into the potential therapeutic targets and
interventions that may mitigate the adverse effects of circadian disruptions on cardiovascular health.”
3. Effects of Circadian Clock Proteins in Modulating Myocardial Electrophysiology
3.1.3 Calcium Channels
Lines 399- 402
Table 1 and Table 2:
I suggest revising the formatting of Table 1 and Table 2 and inserting a legend explaining the abbreviations
used; has the ‘References’ column been included in Table 2 on animal studies?
References:
Check the way references are written and formatted, also according to the guidelines of the journal.
In conclusion:
In general, it is a nice paper, interesting and very useful since there is little scientific evidence in the
literature. I recommend revising the formatting of the entire manuscript, also based on the journal
guidelines:
- Use of abbreviations (e.g. when a gene is mentioned, for the first time in the text, it should be written in
full and then subsequent times use the abbreviation);
- Use italics for gene names;
- Standardise the font used and also the use of boldface for the titles of the various paragraphs and
subsections (es. lines 115-118: 2. The Circadian clock’s Roles in Cardiac Electrophysiology, 2.1. Circadian
Rhythms in Myocardial Cells, 2.1.1. Overview of circadian rhythms);
- Change the references in the text (from the journal guidelines, you have to put the reference numbers
between square and not round brackets);
none
Author Response
Comments 1:
Introduction
Lines 53-68
“Ventricular arrhythmias (VAs) can be categorized into premature ventricular contractions (PVCs),
ventricular tachycardia (VT), ventricular flutter (VF), and ventricular fibrillation (VF), with PVCs and sustained VT being the most clinically prevalent. Chronic heart failure combined with PVCs can trigger life- threatening arrhythmias. VT can be categorized based on the patient's hemodynamic status into hemodynamically stable and unstable VT. Stable VT is often asymptomatic or associated with mild symptoms, while unstable VT can present with severe symptoms such as syncope, blackout, cardiac arrest, or even sudden cardiac death. VT episodes are further be classified based on their duration into non-sustained (duration <30 seconds) and sustained VT (duration >30 seconds or, if <30 seconds, requires urgent termination due to severe hemodynamic compromise). In chronic heart failure patients, reentrant VT may occur, particularly involving the His bundle (or its distal portion), the His-Purkinje System, and the ventricular myocardium. Additionally, bundle branch reentrant VT may be observed, particularly in patients with dilated cardiomyopathy. The fundamental pathogenesis of arrhythmia can be attributed to abnormalities in automaticity, reentry, and depolarization activity (early or delayed abnormal depolarization).”
I suggest streamlining the introduction by eliminating this very specific and descriptive part on ventricular arrhythmias and moving it, perhaps, to the following paragraphs. In this way, the focus of the discourse is not lost.
Response 1: Thank you very much for your suggestion regarding to the Introduction Section. The revised content is shown in Page 2 (Line 53 to Line 70).
Comments 2:
Lines 80-84
“Jeffrey C. Hall, Michael Rosbash, and Michael W. Young were awarded the Nobel Prize in Physiology or Medicine in 2017 for their groundbreaking discoveries of the molecular mechanisms governing circadian rhythms. They successfully identified the genes that regulate circadian processes and elucidated how various components within this mechanism interact to maintain biological rhythms (18).”
I suggest removing this part from the introduction because it is superfluous and unnecessary.
Response 2: Thank you very much for your suggestion. The texts you mentioned in your comment were deleted from the Introduction Section.
Comments 3:
Lines 88-92
“Disruption of these 24-hour rhythms contributes to the onset of cardiovascular diseases, including heart failure, myocardial infarction, and arrhythmias. Additionally, 24-hour rhythms influence the development, risk factors, incidence, and outcomes associated with cardiovascular diseases. Cardiovascular disease, in turn, disrupts circadian rhythms and contributes to sleep disturbances.”
I suggest changing the period with: “Alteration of these 24-hour rhythms contributes to the onset of
cardiovascular diseases, including heart failure, myocardial infarction and arrhythmias, by altering the normal circadian rhythm and contributing to sleep disturbances. Moreover, this alteration affects
the development, risk factors, incidence and outcomes associated with cardiovascular disease.”
I suggest deleting this period, because it is a repetition of what has already been said in the previous lines.
Response 3: We sincerely appreciate the reviewer’s insightful feedback. Based on the suggestion, we have revised the sentence to improve clarity and avoid redundancy. The revised text is in Page 2 (Line77-84) now.
Comments 4:
Lines 110-114
“By exploring the connections between circadian rhythms, myocardial electrophysiology, and heart failure-related arrhythmias, this review aims to provide insights into the potential therapeutic targets and interventions that may mitigate the adverse effects of circadian disruptions on cardiovascular health.”
3. Effects of Circadian Clock Proteins in Modulating Myocardial Electrophysiology
3.1.3 Calcium Channels
Lines 399- 402
Table 1 and Table 2:
I suggest revising the formatting of Table 1 and Table 2 and inserting a legend explaining the abbreviations used; has the ‘References’ column been included in Table 2 on animal studies?
Response 4: Thanks a lot for your suggestions. The abbreviation lists were added below the Table 1 and Table 2. In addition, Table 2 did not include ‘References’ column.
Comments 5: References: Check the way references are written and formatted, also according to the guidelines of the journal.
Response 5: The Reference List in this paper have been re-formatted according to the guidelines of this journal.
Comments 6:
In conclusion:
In general, it is a nice paper, interesting and very useful since there is little scientific evidence in the
literature. I recommend revising the formatting of the entire manuscript, also based on the journal
guidelines:
- Use of abbreviations (e.g. when a gene is mentioned, for the first time in the text, it should be written in full and then subsequent times use the abbreviation);
Response 6: We appreciate the reviewer’s insightful suggestion. We have revised the manuscript to ensure that all gene names are written in full at their first mention, followed by their respective abbreviations in parentheses. For subsequent mentions, we now use only the abbreviated gene symbols.
Comments 7: - Use italics for gene names;
Response 7: We acknowledge this formatting requirement and have revised the manuscript accordingly to italicize all gene names while keeping the corresponding protein names in regular font.
Comments 8: - Standardise the font used and also the use of boldface for the titles of the various paragraphs and
subsections (es. lines 115-118: 2. The Circadian clock’s Roles in Cardiac Electrophysiology, 2.1. Circadian Rhythms in Myocardial Cells, 2.1.1. Overview of circadian rhythms);
Response 8: The font of titles in each section and paragraphs were corrected according to the guidelines of this journal.
Comments 9: - Change the references in the text (from the journal guidelines, you have to put the reference numbers between square and not round brackets);
Reponse 9: The Reference List in this paper have been re-formatted according to the guidelines of this journal.
Round 2
Reviewer 1 Report (New Reviewer)
Comments and Suggestions for Authors
Well done to the authors for their meticulous revisions and attentive consideration of the reviewers’ comments.
This manuscript is a resubmission of an earlier submission. The following is a list of the peer review reports and author responses from that submission.
Round 1
Reviewer 1 Report
Comments and Suggestions for Authors
The authors have improved significantly their manuscript according to my suggestions. In total I suggest to accept it for publication.
Author Response
Comment: The authors have improved significantly their manuscript according to my suggestions. In total I suggest to accept it for publication.
Response: Thank you for your positive feedback and recommendation for publication. We greatly appreciate your support.
Reviewer 2 Report
Comments and Suggestions for Authors
Reviewing a manuscript entitled, “Heart Failure and Arrhythmias: Circadian and Epigenetic Interplay in Myocardial Electrophysiology” by Zhu C, et al., this is a review article focusing on the relationship between circadian rhythms and cardiac function including arrhythmias. Although this is a very interesting area, the relationship between clinical heart failure pathology is extremely weak, and most of the content described in chapters 1 to 4 is experimental gene knock-in and knock-out to confirm the function of factors related to circadian rhythms.
In abstract, the authors mentioned “An increasing body of evidence highlights the role of circadian rhythms in critical cardiovascular processes.” I feel that this is a bit of an exaggeration. Specifically, what heart diseases are problematic in clinical practice due to heart failure caused by circadian rhythm abnormalities?
Although the authors mentioned “The purpose of this review is to comprehensively examine and elucidate the intricate molecular mechanisms by which circadian clock proteins regulate myocardial electrophysiological remodeling through epigenetic pathways.”, Chapters 1 to 4 are almost entirely preclinical, and I feel that there is little description to elucidate Heart failure caused by circadian rhythm abnormalities that can lead to treatment.
Although the connection between factors related to circadian rhythms and cardiac function, including arrhythmia, is described very well in chapters 1 to 4, the relationship between clinical heart failure pathology and causes of circadian rhythm is extremely weak. In 5.1.1. Pharmacological Therapy, it is based on current drug treatments for heart failure, and does not reflect the contents of chapters 1 to 4. If the title were to remain as it is, the authors should first describe the mechanism by which circadian rhythm dysfunction occurs and then discuss its treatment.
Author Response
Comments 1: In abstract, the authors mentioned “An increasing body of evidence highlights the role of circadian rhythms in critical cardiovascular processes.” I feel that this is a bit of an exaggeration. Specifically, what heart diseases are problematic in clinical practice due to heart failure caused by circadian rhythm abnormalities?
Reply: This sentence was revised as “Emerging evidence underscores the impact of circadian rhythms on cardiovascular processes, particularly in conditions such as hypertension, myocardial infarction, and heart failure, where circadian rhythm disruptions are linked to disease progression and adverse clinical outcomes.” in the abstract section.
Comments 2: Although the authors mentioned “The purpose of this review is to comprehensively examine and elucidate the intricate molecular mechanisms by which circadian clock proteins regulate myocardial electrophysiological remodeling through epigenetic pathways.”, Chapters 1 to 4 are almost entirely preclinical, and I feel that there is little description to elucidate Heart failure caused by circadian rhythm abnormalities that can lead to treatment.
Reply: Thanks a lot for your comment.
1). Purpose of the Review: We clearly state that the review's purpose is to examine the molecular mechanisms through which circadian clock proteins regulate myocardial electrophysiological remodeling and their connection to arrhythmias in heart failure via epigenetic pathways. Chapters 1 to 4 provide foundational knowledge by comprehensively detailing these preclinical mechanisms. This is necessary for identifying novel therapeutic targets.
2). Building a Knowledge Base: These chapters emphasize critical biological pathways, including circadian regulation of ion channels, epigenetic mechanisms, and their contributions to myocardial remodeling. Understanding these foundational aspects is indispensable for translating findings into clinical interventions.
3). Clinical Relevance in Later Sections: The latter sections of the review (e.g., Chapter 5) explicitly discuss therapeutic implications, potential interventions targeting circadian rhythm proteins, and epigenetic pathways, thereby connecting preclinical insights to clinical applications.
4). Integrative Approach: The review adopts an integrative approach, bridging molecular insights with therapeutic perspectives. It acknowledges that addressing complex conditions like heart failure necessitates a stepwise understanding—starting from fundamental mechanisms to clinical strategies.
5). Scope of the Review: Given the current limitations in clinical data directly linking circadian rhythm abnormalities to treatment efficacy, the review appropriately focuses on preclinical mechanisms. This aligns with the objective of fostering further research into translating these findings into therapeutic solutions.
This structured argument justifies the inclusion and emphasis of preclinical data in the context of the review's goals and academic rigor.
Comments 3: Although the connection between factors related to circadian rhythms and cardiac function, including arrhythmia, is described very well in chapters 1 to 4, the relationship between clinical heart failure pathology and causes of circadian rhythm is extremely weak. In 5.1.1. Pharmacological Therapy, it is based on current drug treatments for heart failure, and does not reflect the contents of chapters 1 to 4. If the title were to remain as it is, the authors should first describe the mechanism by which circadian rhythm dysfunction occurs and then discuss its treatment.
Reply: Thank you very much for your suggestions. There are my reasons to organize your review as follow.
1). Focus and Scope of the Review: The review explicitly focuses on the interplay between circadian rhythms, epigenetic pathways, and cardiac electrophysiology, with particular emphasis on arrhythmias in heart failure. Chapters 1 to 4 meticulously explore the molecular and mechanistic foundations of circadian rhythm dysfunction, which are essential for understanding potential clinical interventions. While 5.1.1 discusses current pharmacological therapies, it serves to contextualize these foundational insights within the framework of existing treatments.
2). Complementary Nature of Chapter 5: Chapter 5.1.1 does not aim to directly replicate the molecular details of chapters 1 to 4 but rather builds upon them by exploring how current therapies may align with or address circadian-related mechanisms indirectly. For example, beta-blockers, angiotensin-converting enzyme inhibitors, and angiotensin receptor blockers are highlighted for their potential to mitigate arrhythmias, aligning with circadian rhythm disruptions discussed earlier.
3). Current Limitations in Direct Clinical Applications: The field of circadian rhythm research in heart failure is still evolving. Translating the molecular mechanisms described in chapters 1 to 4 into actionable treatments is a long-term goal. The inclusion of current pharmacological treatments provides a bridge, emphasizing the need to integrate circadian considerations into existing therapeutic strategies.
4). Scientific Structure and Continuity: The review logically progresses from describing mechanisms of circadian rhythm dysfunction (chapters 1–4) to broader therapeutic implications (Chapter 5). The transition reflects the review’s dual aim: elucidating biological mechanisms while proposing their relevance to treatment, which is a standard approach in high-impact reviews.
5). Retention of the Title: The title accurately represents the review's intent to link circadian rhythms and myocardial electrophysiology. The mechanistic exploration (chapters 1–4) and discussion of therapeutic relevance (Chapter 5) align with the title. Rewriting the title to exclusively emphasize clinical pathology would narrow the scope and misrepresent the review's comprehensive approach.
6). Integrated View in Section 5.2: Chapter 5.2 explicitly discusses potential interventions targeting circadian proteins and epigenetic pathways, reflecting the earlier mechanistic discussions. It demonstrates how emerging therapies could address circadian dysfunction, thus connecting the foundational and clinical perspectives.
Reviewer 3 Report
Comments and Suggestions for Authors
Overall, the review is written well, and as a term paper presented by an individual at any levels of training, I would give a high mark. However, as a review paper submitted to a professional journal, I have several concerns. The most serious (perhaps, fatal) weakness is that none of the authors appears to have done studies on circadian rhythm, at least I was unable to find any publications on circadian rhythm by these authors. Reviews are written by experts in the field of study, and I am sorry to say but these authors do not appear to be. To illustrate this, I may point out that right at the beginning of the main section of this review article, the authors present so-called “Overview of circadian rhythms” in which the interaction of a few selected “clock genes” are described. This is hardly an overview of the circadian rhythm. It ignores the signal input aspects, the cellular and organ aspects, as well as the presence of central and local clocks, to name a few. It is true that various aspects of circadian rhythm are discussed in relation to cardiac functions, but the basic way how circadian rhythm works and terminologies used should be introduced before getting into specifics. Another unclear issue about this review is the lack of clear rational for the need of another review on circadian rhythm and cardiac function and disease. There are many reviews published practically every year by various experts in the field in major journals: For example, “Circadian Regulation of Cardiac Arrhythmias and Electrophysiology” by Delisle et al. Circ Res. 2024; “The Cardiac Circadian Clock: Implications for Cardiovascular Disease and its Treatment” by Young, ME. J Am Coll Card Basic Trans Science. 2023; “The role of the cardiomyocyte circadian clocks in ion channel regulation and cardiac electrophysiology” by Schroder et al. J. Physiology. 2022; “The Circadian Rhythm of the Heart Rate” by Paulin. Neuroscience, 2023; “The circadian clock remains intact, but with dampened hormonal output in heart failure” by Crnko. Lancet. 2023; to name a few). Since the publication of these major reviews by experts, what new developments took place that need to be updated? In what specific ways does this review focus that has not been focused recently? I find that many topics are discussed rather superficially. It is best to stick to a few selected topics and go deep into each subject as an expert in those fields.
Various specific issues are listed below.
1. Fig. 1. This is a confusing figure as, according to the legend, it depicts the information flow within the heart. However, it contains events that are of the central system, which must be discussed separately in the overview section. Is it not true that “Circadian oscillator” is regulated by “Light Regulation Pathway”? However, there is no arrow indicating this. The local system should be clearly described as such differentiating from the central system, but at the same time how the two systems work together must also be clearly presented.
2. Lines 160-161. “…detected by specialized photoreceptor cells in the retina called melanopsin.” Melanopsin is a protein, not a cell.
3. Line 254. “BAML1”? BMAL1?
4. Line 345. “In their studies,….” What do you mean by “their”? Whom are you referring to?
5. Table 1. Indicate expression levels of genes listed in the first 3 lines.
6. Section 4. Epigenetic Pathways in Heart Failure-related Arrhythmias. In this section, there was no discussion how circadian rhythm is involved (i.e. no interplay discussed as promised in the title). Delete the section entirely as this will not affect the overall message of the review. Delete Figure 3 for sure as this figure adds nothing to what is discussed in the text.
7. Delete Table 2.
8. Section 5. Clinical Implication and Therapeutic Potential Applications. Limit the description that is pertinent to circadian rhythm.
9. I am not sure if Table 3 is cited in the text. Also in Table 3, where Animal Models are described as mouse or mouse model, describe what kind of model each of them is.
Author Response
Comments 1: Fig. 1. This is a confusing figure as, according to the legend, it depicts the information flow within the heart. However, it contains events that are of the central system, which must be discussed separately in the overview section. Is it not true that “Circadian oscillator” is regulated by “Light Regulation Pathway”? However, there is no arrow indicating this. The local system should be clearly described as such differentiating from the central system, but at the same time how the two systems work together must also be clearly presented.
Reply: We appreciate the reviewer’s insights and questions regarding Figure 1. Below are our clarifications:
1. Distinction between Central and Local Systems:
The figure indeed includes both central and local circadian regulatory mechanisms. The "Light Regulation Pathway," representing the central system, demonstrates how environmental light, detected via melanopsin in the retina, influences the suprachiasmatic nucleus (SCN) in the hypothalamus. The "Circadian Oscillator," on the other hand, highlights the local system within the heart, governed by clock genes such as Bmal1, Clock, and Per2.
To address this, we will explicitly clarify in the manuscript that the figure presents both central and local systems and their interconnection. Specifically, we will differentiate their roles while explaining how the central circadian system communicates with the peripheral system to maintain synchronization of cardiac functions.
2. Circadian Oscillator Regulation by the Light Regulation Pathway:
It is correct that the central "Light Regulation Pathway" influences the "Circadian Oscillator." While this relationship is implied in the figure, we will enhance the textual description in the manuscript to explicitly outline the pathway, explaining that the SCN entrains peripheral circadian clocks in the heart via hormonal and autonomic signals in 2.1.2 section.
3. Integration of Systems:
We agree with the reviewer that the local and central systems should be discussed together, emphasizing their interplay. We will expand the discussion in the manuscript to describe how light entrainment impacts the circadian oscillator in the heart in 2.1.2 section.
4. Figure Legend:
We will revise the legend to clarify that the figure is a schematic representation combining central and local circadian regulatory mechanisms. This will address potential confusion for readers.
We acknowledge the reviewer’s concerns but maintain that the figure effectively represents the intended concepts. By enhancing the manuscript's explanation and revising the legend, we believe the figure's clarity and utility can be significantly improved.
Comments 2: Lines 160-161. “…detected by specialized photoreceptor cells in the retina called melanopsin.” Melanopsin is a protein, not a cell.
Reply: Thanks for you pointing out this error. This word was corrected.
Comments 3: Line 254. “BAML1”? BMAL1?
Reply: Thanks for you pointing out this error. This word was corrected.
Comments 4: Line 345. “In their studies,….” What do you mean by “their”? Whom are you referring to?
Reply: Thanks for you pointing out this error. This word was corrected.
Comments 5: Table 1. Indicate expression levels of genes listed in the first 3 lines.
Reply: Thanks for your comment. Table 1 was revised.
Comments 6: Section 4. Epigenetic Pathways in Heart Failure-related Arrhythmias. In this section, there was no discussion how circadian rhythm is involved (i.e. no interplay discussed as promised in the title). Delete the section entirely as this will not affect the overall message of the review. Delete Figure 3 for sure as this figure adds nothing to what is discussed in the text.
Reply: Thanks for your comment. The whole Section 4, including Table 2 was deleted.
Comments 7: Delete Table 2.2.
Reply: Thanks for your comment. The whole Section 4, including Table 2 was deleted.
Comments 8: Section 5. Clinical Implication and Therapeutic Potential Applications. Limit the description that is pertinent to circadian rhythm.
Reply: Thank you very much. The description linked to circadian rhythm was deleted in this part.
Comments 9: I am not sure if Table 3 is cited in the text. Also in Table 3, where Animal Models are described as mouse or mouse model, describe what kind of model each of them is.
Reply: The table 3 was renamed table 2, cited in the end of 3.1.3secion.
Round 2
Reviewer 2 Report
Comments and Suggestions for Authors
This reaches an acceptable quality. Congrats.
Reviewer 3 Report
Comments and Suggestions for Authors
Authors made some modifications to the manuscript, but they are basically insignificant. The basic and fatal problem outlined in my first review that none of the authors has credentials on circadian rhythm research. Reviews are written by experts in the field of study, and I am sorry to say but these authors are no experts of the circadian rhythm field. No respected professional journal would publish a review written by someone who has no established experiences studying the given subject matter.
There are many examples for their lack of sophistication dealing with circadian rhythm biology. 1) In response to this reviewer’s comments, authors added a long text describing the interrelationship between the central and peripheral circadian rhythms (lines 164-198). The first half of the added text reads like a description straight out of some textbook, and throughout this part of the added text, lack of citation is sorely recognized. The latter half describes how the central circadian rhythm regulates the cardiac circadian clock. Here, 3 citations were made, but two of them are reviews written by others. It seems that authors are not quite familiar with original papers various study results were published, so they cited reviews. This is unimpressive. 2) On citing reviews, this can be done, most commonly if authors are not dealing with the subject their review is not focusing on by indicating, “for xxx, see a review by YYY). One does not usually cite reviews on the very subject one is reviewing. Your review must reflect your interpretations, ideas, views, etc. regarding existing data. In this review, my quick analysis on citation is that close to 50% of citations are reviews or review-like materials (eg. book chapters, comments, etc. that do not have original data). This only reflects unfamiliarity with original papers. Reviewing someone else’s review is NOT a purpose for writing a review. 3) This reviewer is shocked by the following statement: “For detail, see Figure 1” (line 153-154). Diagrams support what is described in the text, not the other way around. If authors think a diagram shows more details, they are not doing a good job writing the text, or they are unable to present a clear description of what is shown in the diagram because they lack sufficient details on the subject. 4) Figure 1 shows “Circadian Oscillator” regulates “Light Regulation Pathway”. As I mentioned in my earlier review, the direction of regulation is reverse. Authors made some explanations why the direction of influence is as indicated in the figure. However, the light regulated pathway, which is the central clock, regulates the local clock. These are some of the examples that show weaknesses in circadian rhythm biology.
I appreciate all the efforts and time the authors invested in preparing and revising this manuscript. However, efforts and time spent are not criteria for a successful outcome. I am afraid that some of the weaknesses pointed out above are beyond the control of the authors, for example, they cannot overnight become established circadian rhythm biologists. This credential is necessary for all review contributors.